# Therapeutic Patterns and Clinical Outcomes in Limited Disease Small Cell Lung Cancer: A Decade of Analysis at a Tertiary Cancer Center

**DOI:** 10.3390/cancers16111953

**Published:** 2024-05-21

**Authors:** David Alexander Ziegler, Cecilia Christiane Cleve, Sonia Ziegler, Markus Anton Schirmer, Laura Anna Fischer, Hanibal Bohnenberger, Tobias Raphael Overbeck, Friederike Braulke, Alexander von Hammerstein-Equord, Martin Leu, Manuel Guhlich, Leif Hendrik Dröge, Stefan Rieken, Achim Rittmeyer, Rami A. El Shafie

**Affiliations:** 1Department of Radiotherapy and Radiation Oncology, University Medical Center Göttingen, Robert-Koch-Str. 40, 37075 Göttingen, Germany; cecilia.cleve@stud.uni-goettingen.de (C.C.C.); mschirmer@med.uni-goettingen.de (M.A.S.); laura-anna.fischer@med.uni-goettingen.de (L.A.F.); martin.leu@med.uni-goettingen.de (M.L.); manuel.guhlich@med.uni-goettingen.de (M.G.); hendrik.droege@med.uni-goettingen.de (L.H.D.); stefan.rieken@med.uni-goettingen.de (S.R.); rami.elshafie@med.uni-goettingen.de (R.A.E.S.); 2Göttingen Comprehensive Cancer Center (G-CCC), University Medical Center Göttingen, Von-Bar-Str. 2/4, 37075 Göttingen, Germany; hanibal.bohnenberger@med.uni-goettingen.de (H.B.); friederike.braulke@med.uni-goettingen.de (F.B.); alexander.hammerstein@med.uni-goettingen.de (A.v.H.-E.); a.rittmeyer@lungenfachklinik-immenhausen.de (A.R.); 3Institute of Pathology, University Medical Center Göttingen, Robert-Koch-Str. 40, 37075 Göttingen, Germany; 4Department of Hematology and Medical Oncology, University Medical Center Göttingen, Robert-Koch-Str. 40, 37075 Göttingen, Germany; 5Department of Thoracic and Cardiovascular Surgery, University Medical Center Göttingen, Robert-Koch-Str. 40, 37075 Göttingen, Germany; 6Lungenfachklinik Immenhausen, 34376 Immenhausen, Germany

**Keywords:** small cell lung cancer, cisplatin, carboplatin, etoposide, PCI, LD-SCLC, limited disease lung cancer, outcomes, brain metastases

## Abstract

**Simple Summary:**

The evaluation of therapeutic approaches over a decade at a tertiary cancer center in Germany was undertaken due to the frequent and critical discussions surrounding the selection of chemotherapy regimens and the incorporation of prophylactic cranial irradiation (PCI) in the management of limited disease small cell lung cancer (LD-SCLC). We found significant differences in progression-free survival (PFS) and overall survival (OS) between patients treated with carboplatin and etoposide (CP) and those treated with cisplatin and etoposide (EP), with EP showing better outcomes. Concomitant chemotherapy was also associated with improved PFS and OS. Notably, PCI was found to significantly improve OS and showed a trend towards improved PFS. Female patients had better OS, and the type of chemotherapy and PCI remained independent predictors of survival. This research provides valuable insights into treatment patterns, outcomes, and survival predictors for LD-SCLC, emphasizing the importance of personalized treatment approaches and the potential benefits of PCI. However, the study’s retrospective nature and sample-size limitations should be considered, highlighting the need for further prospective studies in this field.

**Abstract:**

In this study, we investigated the outcomes and factors influencing treatment efficacy in 93 patients with limited disease small cell lung cancer (LD-SCLC), with a median age of 64 years. We focused on the impact of chemotherapy regimens, prophylactic cranial irradiation (PCI), and patient-related variables. The median follow-up for OS was 17.3 months. We observed a statistically significant difference in PFS between LD-SCLC patients treated with cisplatin and etoposide (EP) and those treated with carboplatin and etoposide (CP) (PFS: EP 13.63 months vs. CP 6.54 months, *p* < 0.01). Patients treated with EP had better overall survival (OS) than CP-treated patients (OS: EP 26.9 months vs. CP 16.16 months, *p* < 0.01). Concomitant chemotherapy was associated with improved PFS (*p* = 0.003) and OS (*p* = 0.002). Patients receiving PCI showed superior OS (*p* = 0.05) and a trend towards improved PFS (*p* = 0.057). Female gender was associated with better OS (*p* = 0.025). Most patients had an ECOG performance status of 0 (71%). This real-world study underscores the importance of multidisciplinary LD-SCLC management, emphasizing the roles of chemotherapy, radiotherapy, and PCI. These findings inform personalized treatment strategies and emphasize the need for prospective trials to validate these results and optimize LD-SCLC treatment.

## 1. Introduction

Small cell lung cancer (SCLC) accounts for approximately 10–15% of all lung cancer cases. It is characterized by high proliferation rates, rapid dissemination into locoregional lymph nodes, and the formation of distant metastases [1]. Limited disease SCLC (LD-SCLC) accounts for approximately 40% of initial diagnoses and, in those cases, a multidisciplinary treatment consisting of chemotherapy (CT) and radiotherapy (RT) and, in very early stages, surgery, is indicated [2]. CT is commonly applied in at least four courses of a combination regimen consisting of either cisplatin and etoposide (EP) or carboplatin and etoposide (CP) [3]. Thoracic RT is optimally delivered in twice-daily fractions of 1.5 Gy to a total of 45 Gy [4] or even up to 60 Gy [5]. Alternatively, a once-daily regimen of 1.8–2.0 Gy to a total dose of 60–70 Gy is employed [6].

A randomized clinical trial of the European Organization for Research and Treatment of Cancer (EORTC) showed that, after the initial response to first-line chemotherapy, prophylactic cranial irradiation (PCI) reduced the risk of brain metastases and induced a survival improvement in patients with LD-SCLC [7] and extensive disease SCLC (ED-SCLC) [8]. A Japanese study reported that close surveillance with regular magnetic resonance imaging (MRI) as an alternative to PCI led to similar survival results and a better quality of life in patients with ED-SCLC [9].

As shown above, a relative inhomogeneity exists in applied treatment strategies regarding the thoracic, as well as the cerebral component of LD-SCLC. Prospective clinical trials, on which the existing evidence is based, typically include a selected patient collective and clinicians often face difficulties in applying resulting recommendations to their patients, depending on overall clinical performance and co-morbidities. This study aims to gather real-world evidence for patients with LD-SCLC patients treated at a lung cancer center certified according to the requirements of the German Cancer Society (Deutsche Krebsgesellschaft, DKG) in a large tertiary care German cancer center over a ten-year period.

## 2. Patients and Methods

### 2.1. Cohort Description

Patients were eligible for inclusion if they underwent both chemotherapy and thoracic radiotherapy (CRT) with either EP (etoposide and cisplatin) or CP (carboplatin and etoposide). The exclusion criteria encompassed cases with mixed histology, those receiving mixed or no chemotherapy concurrent with or prior to radiotherapy, and patients who had undergone surgical resection of the tumor or its parts, with the exception of a biopsy before radiotherapy.

Very limited stage cancer is defined by the Veterans Administration Lung Study Group [10,11] as a localized tumor disease without mediastinal lymph node metastases, corresponding to a classification in the TNM stage of either T1 or T2 and N0 or N1. Limited stage was defined as cancer summary stage I-III by the Veterans Administration Lung Study: a tumor that is initially localized to one side of the chest, with or without the presence of metastases in the mediastinal or supraclavicular lymph nodes on the same side (ipsilateral) or the opposite side (contralateral), and may or may not be accompanied by a pleural effusion on the same side, regardless of the cytological test results. TNM classification was based on currently applicable guidelines at the time of diagnosis.

### 2.2. Radiotherapy Target Volume Delineation and Treatment Planning

Radiotherapy was applied at a linear accelerator with daily image guidance and utilizing photons at 6 MeV or 20 MeV. Three-dimensional conformal radiotherapy (n = 63) and volumetric modulated arc therapy (Rapid-Arc; n = 30) were used for RT. For dose calculation, the Eclipse treatment planning system (Varian Medical Systems, Palo Alto, CA, USA) was used. The treatment planning CT was co-registered with contrast-enhanced CT and PET-CT, if available. The Gross Tumor Volume of the primary tumor (GTVp) encompassed all anatomically visible tumor tissue associated with the primary tumor, and the Gross Tumor Volume of affected lymph nodes (GTVn) encompassed all macroscopic lymph nodes with suspected tumor involvement. The clinical target volume (CTV) was based on applicable guidelines at the time of RT and adapted to anatomical boundaries. Initially (pre-chemotherapy) the affected lymph node levels and, in all cases, the ipsilateral hilus (level 10) and ipsilateral mediastinal lymph node levels 4 and 7 were defined as the nodal/mediastinal clinical target volume (CTVn). If anatomically reasonable, a connection between CTVn and CTVp was added. Planning target volume (PTV) was defined as combined CTVp and CTVn with a safety margin of 10 mm in all directions.

According to the German guidelines, prophylactic cranial irradiation (PCI) was routinely offered to patients who responded to radiochemotherapy. For this purpose, a CT scan of the chest had to be performed 4–6 weeks after thoracic radiotherapy, showing remission. An MRI of the head was only required before the initiation of thoracic radiotherapy, with no existing cerebral metastases. PCI was administered in either 10 fractions of 2.5 Gy each or 15 fractions of 2 Gy each.

### 2.3. Chemotherapy

All patients received either EP or CP. A change in chemotherapy regimen during ongoing treatment led to exclusion from this analysis. EP was given every 3 weeks (4 cycles) at a cisplatin dose of 60 mg/m^2^ on day 1 and etoposide at a dose of 120 mg/m^2^ on days 1, 2, and 3. CP was given every 3 weeks (4–6 cycles), with carboplatin (AUC 5) on day 1 and etoposide at a dose of 100 mg/m^2^ on days 1, 2, and 3 [12,13]. The decision between EP and CP was made by the treating oncologist within the lung tumor board mostly based on kidney function and other co-morbidities.

### 2.4. Toxicity Scoring and Follow-Up

During RT, patients received a scheduled physician visit at least once weekly and additional visits as required. Toxicity was recorded according to the Common Terminology Criteria for Adverse Events (CTCAE) scoring system [14] at the time data were collected, and, if toxicities > grade 2 occurred, admission to our in-patient ward was evaluated. New toxicities were classified as acute and/or treatment-associated toxicities if they occurred within three months after RT and late toxicities if they occurred later. After the completion of CRT, all patients were followed up regularly according to applicable guidelines until tumor progression or death or the completion of a maximum of 5 years without tumor progression.

### 2.5. Statistical Analyses

Baseline analyses are reported descriptively using mean (with standard deviation SD), where appropriate, or median (quartiles, range) and categorical variables as absolute and relative frequencies. Overall survival (OS) was calculated from the date of first irradiation until death or censored at the last observation. Progression-free survival (PFS) was calculated from the date of first irradiation until tumor progression, death, or the last observation. The median follow-up time was calculated using the reverse Kaplan–Meier method [15]. To identify prognostic factors on PFS and OS, univariable and multivariable Cox regression analyses were used. Variables with statistical significance in univariable analysis underwent multivariable modeling (applicable to all patients) and were considered in multivariable Cox regression analyses. For highly correlated variables, the one with the most presumed clinical relevance was included in the multivariable Cox regression. Due to the retrospective nature of this study, *p*-values are of a descriptive nature. Descriptive *p*-values < 0.05 were considered statistically significant. All statistical analyses were performed using IBM SPSS Statistics, Version 28 (New York, NY, USA). This retrospective study was approved by the Ethics Committee of the Medical Faculty of Göttingen (ref. no.: 9/4/21).

## 3. Results

### 3.1. Patient Cohort and Characteristics

Between 2007 and 2018, 486 patients were identified who were diagnosed with SCLC and received radiotherapy within our department and discussed their condition with a multi-professional team of experts within the interdisciplinary tumor board of the certified lung cancer center Göttingen. Out of the 486 patients with SCLC, 276 patients (61%) had ED-SCLC at initial diagnosis. For 178 patients (36%), LD-SCLC or very limited disease SCLC (VLD-SCLC) was documented with uniform histology (no mixed tumor cell histology) defined through pathological examination and radiotherapy was administered. Of these 178 patients, 146 had LD-SCLC (82%) and 32 patients (18%) had VLD-SCLC.

According to our pre-specified exclusion criteria, of the considered 178 patients with either LD-SCLC or VLD-SCLC, those were excluded who received no CP or EP chemotherapy regimen (n = 75; 22%), thoracic resection before RT (n = 5; 2.8%), or who were treated only with PCI (n = 5; 2.8%). Finally, 93 patients were included in this study (VLD-SCLC n = 15; 16.1% and LD-SCLC n = 78; 83.9%). Details on inclusion and exclusion of patients are illustrated in the flow diagram (Figure 1).

The median patient age at the time of first RT was 65 years (Q1–Q3: 58–70 years).

ECOG performance status was predominantly 0 or 1 (ECOG 0: n = 66, 71%; ECOG 1: n = 24, 25.8%, ECOG 2: n = 2, 2.2%; ECOG 4: n = 1, 1.1%). When categorizing ECOG performance status into 0–1 vs. ≥2, it becomes evident that, in the CP cohort, one patient falls into the ECOG ≥ 2 group (3.2%), and, in the EP cohort, two patients exhibit ECOG ≥2 (3.2%). No statistical differences were observed between these two groups. Upon stratifying ECOG as 0 vs. ≥1, in the CP cohort, there were 16 patients (51.6%) vs. 15 (48.4%), and, in the EP cohort, there were 50 (80.6%) vs. 12 (19.4%). These differences were statistically significant (*p* = 0.04).

The age-adjusted Charlson Comorbidity Index (CCI) [16] of the patients ranged from 0 to 8 (CCI 0: n = 3, 3.22%; CCI 1: n = 14, 15.1%; CCI 2: n = 23, 24.7%; CCI 3: n = 28, 30.1%; CCI 4: n = 14, 15.1%; CCI 5: n = 8, 8.6%; CCI 6: n = 2, 2.2%; CCI 8: n = 1, 1.1%). Out of the 31 patients receiving CP, 6 (19.4%) had a CCI of ≥5. Meanwhile, 5 out of 62 patients (8.1%) receiving EP had a CCI of ≥5. These differences were not significant based on the chi-squared test (*p* = 0.11).

When considering the size of the target volumes, the CP cohort exhibited a median CTV volume of 216.6 cm^3^ (min 56.9 cm^3^; max 468 cm^3^) and a median PTV volume of 623.7 cm^3^ (min 207 cm^3^; max 1187.2 cm^3^). In contrast, the EP cohort had a median CTV volume of 178.5 cm^3^ (min 15.2 cm^3^; max 901.7 cm^3^) and a median PTV volume of 500.7 cm^3^ (min 83.3 cm^3^; max 1467.1 cm^3^). These differences were not significant (*p* = 0.53) based on the Kruskal–Wallis–Test.

As by TNM stage, the EP cohort showed the following distribution of patients with T classification: 4 (6.5%) at stage cT1, 21 (33.9%) at stage cT2, 10 (16.1%) at stage cT3, and 27 (43.5%) at stage cT4. In the CP cohort, patients were distributed in T classification as follows: 6 (19.4%) at stage cT1, 6 (19.4%) at stage cT2, 4 (12.9%) at stage cT3, and 15 (48.4%) at stage cT4. These differences were not significant (*p* = 0.9) based on the Kruskal–Wallis–Test.

Regarding lymph node status, the EP cohort showed the following distribution of patients: 10 (16.1%) at stage cN0, 8 (12.9%) at stage cN1, 27 (43.5%) at stage cN2, and 16 (25.8%) at stage cN3. In the CP cohort, patients were distributed based on lymph node status as follows: 9 (29.0%) at stage cN0, 6 (19.4%) at stage cN1, 7 (22.6%) at stage cN2, and 9 (29.0%) at stage cN3. These differences were not significant (*p* = 0.34) based on the Kruskal–Wallis–Test.

Patients were divided into two cohorts depending on the chemotherapy applied—characteristics are illustrated in Table 1.

All patients received thoracic RT at a dose of from 45 to 60 Gy (n = 93) and 78 (83.9%) patients received additional prophylactic cranial irradiation (PCI). A total of 49 (52.7%) patients were treated bi-daily at a single dose of 1.5 Gy based on the Turrisi protocol [4] and 44 (47.3%) patients were treated once daily at a single dose of from 1.8 Gy to 2 Gy with a cumulative total dose between 45 and 60 Gy.

Three patients receiving EP did not receive the prescribed RT dose because of toxicity (3/62 = 4.8%).

Within the CP cohort, 8/31 patients (25.8%) received systemic therapy concomitant with radiotherapy and, within the EP cohort, 45/62 patients (72.6%) received systemic therapy concomitant with radiotherapy (please see Table 2 for further details). In the remaining cases, treatment was combined sequentially, applying RT after chemotherapy completion.

Among the 62 patients receiving EP, 16 (25.8%) experienced side effects or hematotoxicity of grade ≥ III, while, in the CP (n = 31) cohort, 9 (29%) had side effects or hematotoxicity of grade ≥ III (Table 1). These differences did not reach statistical significance in the chi-squared test.

### 3.2. Overall Survival (OS)

The median follow-up for OS, as estimated by the reverse Kaplan–Meier method, was 17.3 months (Q1–Q3: 9.6–73.7; 95%-CI: 12–22.7). The median OS for the CP group was 16.16 months (Q1–Q3: 11.60–24.74; 95%-CI: 13.15–19.18), and, for the EP group, it was 26.94 months (Q1–Q3: 16.88–96.43; 95%-CI: 5.26–48.61), (HR 0.49, 95%-CI: 0.28–0.75, *p* = 0.006; Figure 2). OS after 12 months was 67.1% for the CP cohort and 82.0% for the EP cohort. In the univariable Cox analysis (Table 3), concomitant chemotherapy was predictive of longer OS (HR 0.46, 95%-CI: 0.28–0.75, *p* = 0.002). These patients had a median OS of 48.46 months (Q1–Q3: 13.5–21.86; 95%-CI: 21.81–75.11), while those without concomitant chemotherapy exhibited a median OS of 18.43 months (Q1–Q3: 15.15–24.74; 95%-CI: 16.4–20.5). The female gender showed a significant survival advantage over the male (HR 0.58, 95%-CI: 0.36–0.93, *p* = 0.025).

This study revealed a significantly superior OS for the PCI group compared to patients not receiving PCI (HR 0.53, 95%-CI: 0.28–0.98, *p* = 0.04; Figure 3). The median OS for patients who received PCI was 23.7 months (Q1–Q3: 15.7–75.8; 95%-CI = 3.9–31.3), while the non-PCI group’s median OS was 13.1 months (Q1–Q3: 6.4–26.9; 95%-CI = 5.8–27.5).

Stratifying by tumor stage revealed 15 (16.1%) with VLD and 78 (83.9%) with LD. The median OS was 27.60 months for VLD (IQR: 12.75–75.27; 95%-CI: 15.56–39.64) and 19.29 months for LD (IQR: 13.83–24.95; 95%-CI: 17.31–21.27), with no significant difference in OS (HR 1.17, 95%-CI: 0.64–2.12, *p* = 0.611).

In the multivariable analysis, female sex (HR 0.49, 95%-CI: 0.29–0.80, *p* = 0.004), PCI (HR 0.27, 95%-CI: 0.13–0.54, *p* < 0.001), simultaneous radio-chemotherapy (HR 0.43, 95%-CI: 0.24–0.77, *p* = 0.004), and EP (HR 0.57, 95%-CI: 0.32–0.99, *p* = 0.049) were associated with superior OS (Table 4).

Comparing patients with definitive SCLC histology who were not included in this study (e.g., due to a change in chemotherapy or thoracic tumor resection; n = 85) with those who were included (n = 93), there were no statistically significant differences in OS (*p* = 0.52) (Appendix A). The median OS for the included patients was 19.8 months (Q1–Q3: 13.3–64.7; 95%-CI: 15.3–24.4), while the median OS for the excluded patients was 20.85 months (Q1–Q3: 12.8–56.5; 95%-CI: 16.9–24.8).

### 3.3. Progression-Free Survival (PFS)

The median PFS in the CP group was 6.54 months (Q1–Q3: 3.19–13.14; 95%-CI: 4.62–8.46), while, for the EP group, it was 13.63 months (Q1–Q3: 6.6–73.69; 95%-CI: 9.75–17.5), resulting in a statistically significant difference between groups (*p* < 0.01; Figure 4). PFS after 12 months among the CP cohort was 30.6%, and, among the EP cohort, it was 57.2%. Of the total, 53 patients (57%) were treated with concomitant chemotherapy, while 40 patients (43%) underwent chemotherapy either before or after RT.

In the univariable analysis, treatment with concomitant chemotherapy was significantly associated with a PFS benefit (HR: 0.49, 95%-CI: 0.31–0.78, *p* = 0.003). Those receiving concomitant chemotherapy had a median PFS of 16.7 months (Q1–Q3: 6.13–74.35; 95%-CI: 8.11–25.33). In contrast, patients not treated with concomitant chemotherapy had a median PFS of 8.28 months (Q1–Q3: 5.09–13.37; 95%-CI: 1.89–14.67). A trend towards improved PFS (Figure 5) in the PCI group was evident (HR: 0.56; 95%-CI: 0.31–1.02; *p* = 0.053), demonstrating a median PFS of 10.3 months (Q1–Q3: 6.1–45.9; 95%-CI: 10.3–15.9) compared to 3.9 months (Q1–Q3: 2–13.6; 95%-CI: 1.5–6.3) in the non-PCI group. Stratifying by tumor stage revealed 15 (16.1%) with VLD and 78 (83.9%) with LD. The median PFS in the VLD stage was 20.34 months (Q1–Q3: 8.48–16.33; 95%-CI: 2.44–38.24), and, in the LD stage, it was 10.25 months (Q1–Q3: 4.99–18.79; 95%-CI: 5.52–14.98). This difference was not statistically significant (HR: 1.55; 95%-CI: 0.86–2.79; *p* = 0.14). Simultaneous radio-chemotherapy (HR 0.49, 95%-CI: 0.31–0.78, *p* = 0.003) and EP (HR 0.42, 95%-CI: 0.26–0.67, *p* = 0.001) were associated with superior PFS (Table 3). In the multivariable analysis, only EP vs. CP stayed independently prognostic of superior PFS (HR 0.50, 95%-CI: 0.29–0.86, *p* = 0.013) (Table 4).

## 4. Discussion

### 4.1. Treatment Patterns and Outcomes

Prior research [12,13] has demonstrated the survival benefits of CRT for (V)LD-SCLC. In this study, we present results that are consistent with these findings. Notably, patients receiving EP exhibited significantly improved PFS and OS compared to those receiving CP. This difference could be attributed to various factors, including patient selection, underlying comorbidities, and potential variations in tumor response. Considering that EP is per protocol given in conjunction with radiotherapy, there is a clear association between EP therapy and simultaneous chemotherapy. While some patients received CP with radiotherapy, this was not the typical approach (see Table 1—Chemotherapy regimen). As per the protocol, a twice-daily radiation treatment following Turrisi was almost exclusively performed on patients with EP, showing significant differences between both cohorts in terms of the total dose and individual fractions. The analysis reveals a significant difference in ECOG performance status between the CP and EP groups. A higher percentage of CP patients had an ECOG score of ≥1, which signifies a poorer performance status. We chose to divide the ECOG performance status into 0 vs. ≥1 as the performance status is likely one of the factors influencing the treating oncologist’s choice between chemotherapy regimens. It was demonstrated that there were significant differences between the cohorts in this regard. This underscores the importance of considering a patient’s baseline health and performance status when selecting treatment options, as it can influence treatment tolerability and outcomes. Although the CCI difference between the CP and EP groups was not statistically significant, it is still a relevant factor in treatment decisions.

There is ongoing uncertainty about the benefits of chemotherapy for elderly lung cancer patients, who are more prone to treatment-related toxicities like myelosuppression due to comorbidities. Geriatric assessment should steer treatment choices, with combination chemotherapy, especially carboplatin and etoposide, being standard for elderly SCLC patients. Studies suggest that lower doses in elderly patients yield comparable outcomes to those in younger patients, and single-agent therapy might provide similar benefits with fewer side effects for those with additional health issues [17]. In a study examining very elderly patients (80–92 years) with SCLC, survival outcomes were notably influenced by the stage of cancer and the type of treatment administered. Particularly, patients who received a combination of chemotherapy and local therapy demonstrated improved survival, illustrating the potential benefits of aggressive, yet carefully considered, treatment strategies for this age group [18]. In a study on ED-SCLC, the effectiveness and safety of a CP regimen were compared with split doses of EP in elderly or high-risk patients. The findings revealed no significant differences in response rates and overall survival between the two treatment regimens, suggesting that the CP regimen could serve as a viable alternative to the conventional EP regimen, especially considering its similar risk–benefit profile [19]. This study demonstrates that, particularly in elderly patients who may have multiple comorbidities and potentially poorer ECOG performance status, the choice of chemotherapy needs to be carefully considered. This emphasizes the importance of tailoring treatment strategies to individual patient conditions to optimize outcomes and minimize risks.

### 4.2. Role of Prophylactic Cranial Irradiation (PCI)

The addition of PCI in the treatment regimen of LD-SCLC has long been debated. The randomized clinical trials conducted by the European Organization for Research and Treatment of Cancer (EORTC) demonstrated the efficacy of PCI in reducing the risk of brain metastases and improving survival [7,8]. In line with these trials, the present study showed that patients who received PCI had a statistically significant improvement in OS, with a trend towards improved PFS.

However, the decision to administer PCI should be made carefully, considering the potential impact on patients’ quality of life. A Japanese study indicated that regular magnetic resonance imaging (MRI) surveillance might be an alternative to PCI, providing comparable survival outcomes and improved quality of life for patients with ED-SCLC [9]. To achieve the best treatment for patients with newly diagnosed brain metastases, current research is investigating whether stereotactic RT of the brain metastases could serve as an alternative with reduced toxicity, as opposed to whole brain irradiation [20,21]. Following the publication of the phase 3 clinical trial by Takahashi et al., which found no OS benefit for PCI in patients with ED-SCLC, the current recommendation for LD-SCLC remains unclear. A Chinese retrospective study compared PCI and active surveillance in patients with LD-SCLC, revealing that while PCI significantly reduced the incidence of brain metastases it did not extend overall survival compared to surveillance [22]. Whether or not MRI surveillance can be an alternative to PCI in this patient cohort remains to be shown in a prospective clinical trial. The EORTC trial, PRIMALung, is currently recruiting participants to address this question [23], as well as a multicentric study initiated by the Shandong Cancer Hospital and Institute [24].

### 4.3. Survival Predictors

This study’s multivariable Cox analysis identified several factors associated with survival outcomes. Female patients demonstrated better overall survival, aligning with previous observations suggesting potential gender-related differences in SCLC outcomes [25]. The analysis also highlighted the importance of the treatment regimen, with patients receiving EP exhibiting improved survival compared to those receiving CP. A possible reason why the EP regimen demonstrates a survival benefit compared to CP in LD-SCLC lies in the differing molecular mechanisms of the platinum compounds used. Cisplatin, utilized in EP, is highly effective due to its robust ability to form both intrastrand and interstrand DNA cross-links, crucial for inhibiting DNA synthesis and function [26]. This capability results in significant cell cycle arrest and apoptosis, particularly advantageous in SCLC which has a high mitotic index and is thus more vulnerable to DNA disruption. In contrast, carboplatin—used in CP—forms similar DNA adducts but with less molecular potency and a slower kinetic profile, making it less effective but also less toxic [27,28,29]. In a meta-analysis, it was shown that patients who received chemotherapy with CP experienced higher hematotoxicity, while other toxicities, such as nausea/vomiting, neurotoxicity, and renal toxicity, were increased in therapies using EP [30]. This reduced toxicity, while beneficial for patient tolerability, particularly in those with comorbidities or compromised organ function, may contribute to a diminished therapeutic impact compared to cisplatin, influencing overall survival outcomes.

Furthermore, this study reinforced the positive impact of PCI on survival, emphasizing its potential role as an important adjunct to CRT for LD-SCLC. The utilization of PCI, along with its potential to reduce the risk of brain metastases, remains an important consideration in clinical decision-making.

Nevertheless, the results also indicate that age and the distinction between stages (VLD or LD) do not serve as predictors for improved OS or PFS. Therefore, when making treatment decisions, it is advisable to take into account the patient’s ECOG performance status instead of the CCI.

### 4.4. Limitations

Being retrospective in nature and comprising a single-center analysis, this study is subject to selection bias and lacks the controlled environment of a randomized trial. The heterogeneous patient population, varied treatment protocols, and potential unmeasured confounding variables might influence the results. Additionally, the relatively small sample size, particularly in the subgroup analysis, limits the generalizability of the findings, and the relatively short follow-up carries a certain risk of overestimating OS due to early censoring in some subgroups.

The restriction that only patients who received either EP or CP were included, and that a change in the chemotherapy regimen led to exclusion from the study, was necessary to compare the effects of chemotherapy with one another. However, this could also potentially introduce a confounder in the analysis. The cohort size was too small to capture and evaluate these patients in a targeted manner. However, in terms of overall survival, there were no differences between the cohort with chemotherapy switch (as well as other factors such as prior surgery) and the patients included in this study.

### 4.5. Strength

When considering real-world data, as opposed to the controlled environments of clinical trials with predetermined schedules, we encounter diverse elderly patient groups facing the complexities of comorbidities and accommodating individual patient preferences in both everyday inpatient and outpatient settings.

This underscores the essential role of multidisciplinary teams within specialized centers. Such teams are vital for crafting personalized, guidelines-driven, patient-centered treatment plans tailored to the unique needs of each individual patient.

## 5. Conclusions

This study provides real-world evidence on treatment patterns, outcomes, and factors associated with survival in patients with LD-SCLC. The results underscore the importance of multidisciplinary treatment approaches, emphasizing the role of chemotherapy, radiotherapy, and the potential benefit of PCI in improving survival outcomes. Nevertheless, the individualized nature of LD-SCLC management requires careful consideration of patients’ characteristics, treatment preferences, and potential toxicities. Future prospective studies and clinical trials will be crucial to validate these findings and further refine treatment strategies for this challenging disease entity.

## Figures and Tables

**Figure 1 cancers-16-01953-f001:**
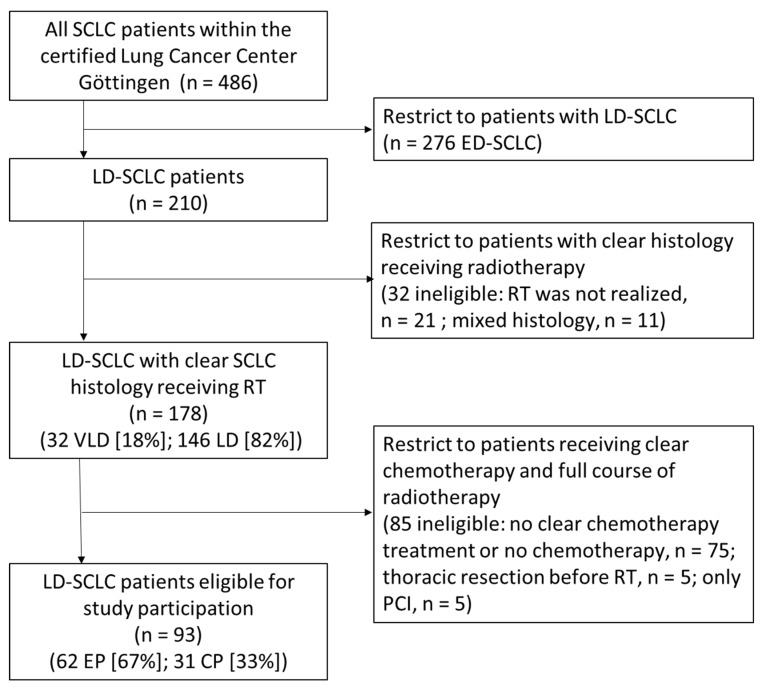
Consort flow diagram; SCLC, small cell lung cancer; RT, radiotherapy; LD-SCLC, limited disease small cell lung cancer; ED-SCLC, extensive disease small cell lung cancer; VLD, very limited disease small cell lung cancer; RCT, radio-chemo-therapy; PCI, prophylactical cranial irradiation; EP, cisplatin/etoposide; CP, carboplatin/etoposide; clear chemotherapy means no change of regimen during treatment; clear histology means no mixed tumor cell histology.

**Figure 2 cancers-16-01953-f002:**
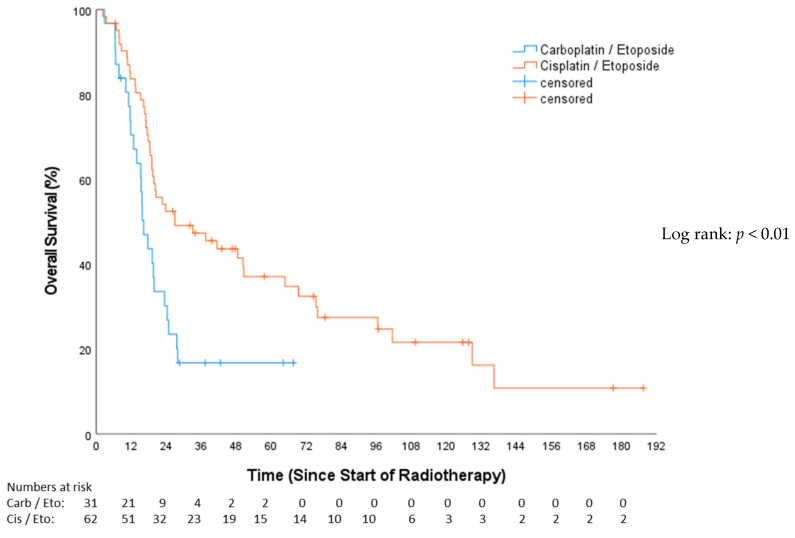
Overall survival—carboplatin/etoposide vs. cisplatin/etoposide.

**Figure 3 cancers-16-01953-f003:**
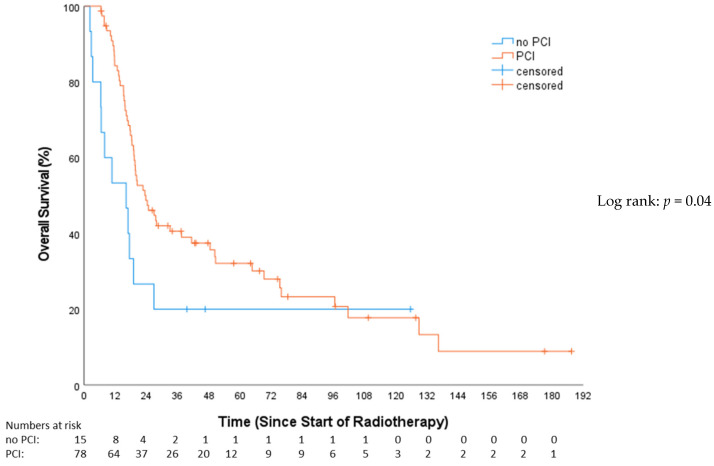
Overall survival—no prophylactical cranial irradiation (no PCI) vs. prophylactical cranial irradiation (PCI).

**Figure 4 cancers-16-01953-f004:**
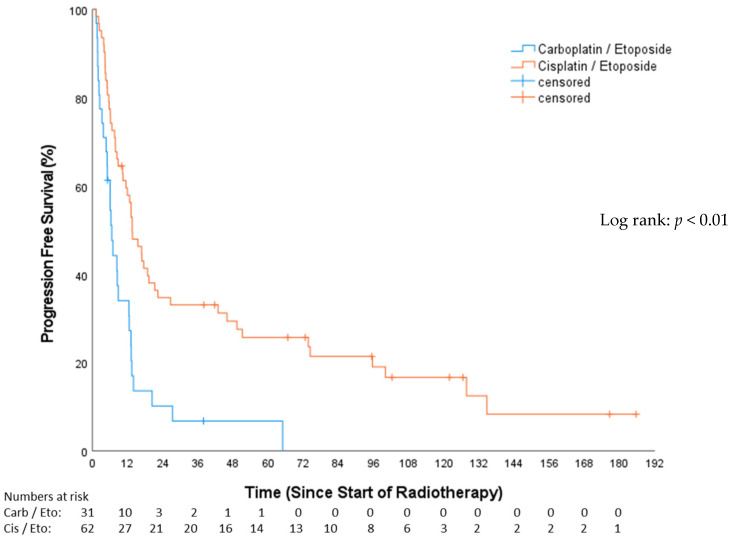
Progression-free survival—carboplatin/etoposide vs. cisplatin/etoposide.

**Figure 5 cancers-16-01953-f005:**
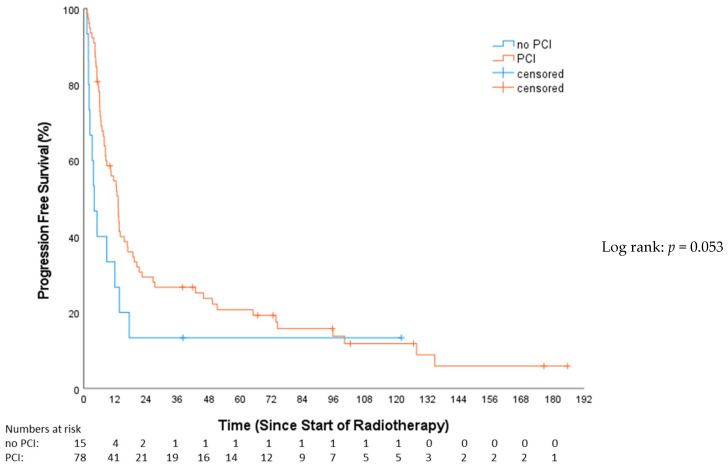
Progression-free survival—no prophylactical cranial irradiation (no PCI) vs. prophylactical cranial irradiation (PCI).

**Table 1 cancers-16-01953-t001:** Patient characteristics.

Variable	EP Cisplatin/Etoposide (%)	CP Carboplatin/Etoposide (%)	*p* Value *
Sex			
male	34 (54.8)	16 (51.6)	
female	28 (45.2)	15 (48.4)	0.77
Age, y			
46–59	27 (43.5)	5 (16.1)	
≥60	35 (56.5)	26 (83.9)	0.01
Stage			
VLD	9 (14.5)	6 (19.4)	
LD	53 (85.5)	25 (80.7)	0.7
Tumor status			
cT1	4 (6.5)	6 (19.4)	
cT2	21 (33.9)	6 (19.4)	
cT3	10 (16.1)	4 (12.9)	
cT4	27 (43.5)	15 (48.4)	0.9
Nodal status			
cN0 cN1 cN2 cN3	10 (16.1) 8 (12.9) 27 (43.5) 16 (25.8)	9 (29.0) 6 (19.4) 7 (22.6) 9 (29.0)	0.34
CTV			
Mean (SD) Min-max	177.3 (148.8) 15.2–901.7	216 (120.9) 56.9–468	0.53
RT twice daily			
yes	40 (64.5)	8 (25.8)	
no	22 (35.5)	23 (74.2)	<0.01
RT dose			
45 Gy twice daily ≤50 Gy once daily	40 (64.5) 16 (25.8)	8 (25.8) 21 (67.7)	
>50 Gy once daily	6 (9.7)	2 (6.4)	<0.01
RT technique			
3D-conformal RT	41 (66.1)	22 (71)	
Rapid-Arc	21 (33.9)	9 (29)	0.64
PCI			
yes no	55 (88.7) 7 (11.3)	27 (87.1) 4 (12.9)	0.55
Smoking status pre RT			
yes	59 (95.2)	29 (93.5)	
no	3 (4.8)	2 (6.5)	0.76
Grade ≥III side effects/hematotoxicity			
yes	16 (25.8)	9 (29)	
no	46 (74.2)	22 (71)	0.74
Eastern Cooperative Oncology Group performance status			
0	50 (80.6)	16 (51.6)	
1–4	12 (19.4)	15 (48.4)	0.04
Charlson comorbidity index			
0–4	57 (91.9)	25 (31)	
≥5	5 (8.1)	6 (19.4)	0.11

EP, cisplatin/etoposide; CP, carboplatin/etoposide; VLD, very limited disease; LD, limited disease; RT, radiotherapy; CTV, clinical target volume; mean total CTV, mean CTV of both cohorts; SD, standard deviation; Gy, gray; PCI, prophylactical cranial irradiation; * *p*-values were determined using the chi-square test for two variables, and the Kruskal–Wallis test for more than two variables.

**Table 2 cancers-16-01953-t002:** Chemotherapy regimen.

	EP Cisplatin/Etoposide (%)	CP Carboplatin/Etoposide (%)
Patients with pre-RT systemic therapy (sequential regimen)	17/62 (27.4%)	23/31 (74.2%)
Patients with systemic therapy concomitant to RT (simultaneous regimen)	45/62 (72.6%)	8/31 (25.8%)
Chemotherapy-cycles pre-RT		
0 cycles	27 (43.5%)	2 (6.5%)
1 cycle	8 (12.9%)	6 (19.4%)
2 cycles	7 (11.3%)	3 (9.7%)
3 cycles	3 (4.8%)	1 (3.2%)
4 cycles	4 (6.5%	8 (25.8%)
5 cycles	1 (1.6%)	1 (3.2%)
6 cycles	11 (17.7%)	10 (32.3%)
Chemotherapy cycles concomitant to the RT		
0 cycles	18 (29%)	23 (74.2%)
1 cycle	31 (50%)	4 (12.9%)
2 cycles	13 (21%)	4 (12.9%)

RT, radiotherapy; EP, cisplatin/etoposide; CP, carboplatin/etoposide.

**Table 3 cancers-16-01953-t003:** Univariable Cox proportional hazard-censored regression analysis of factors associated with overall survival and progression-free survival in patients with limited disease small cell lung cancer.

Variable	Overall Survival			Progression-Free Survival		
	HR	95%-CI for HR	*p* Value	HR	95%-CI for HR	*p* Value
Sex	0.58	0.36–0.93	**0.025**	0.73	0.47–1.14	0.171
Age ≥ 60 y	1.93	0.71–1.93	0.545	1.40	0.87–2.27	0.167
ECOG 0 vs. 1–4	0.45	0.73–2.05	1.223	1.49	0.93–2.38	0.106
Charlson ≥5	1.00	0.46–2.20	0.998	1.31	0.67–2.57	0.425
PCI	0.53	0.28–0.98	**0.** **044**	0.56	0.31–1.02	0.057
Simultaneous RCT	0.46	0.28–0.75	**0.** **002**	0.49	0.31–0.78	**0.** **003**
EP vs. CP	0.49	0.29–0.81	**0.** **006**	0.42	0.26–0.67	**0.** **001**
Hematotoxcity ≥ 3	0.62	0.35–1.08	0.093	0.74	0.45–1.23	0.249
CTV	1.05	0.66–1.68	0.835	1.15	0.74–1.79	0.530
Tumor-Status (T)	1.18	0.93–1.49	0.180	1.25	0.99–1.56	0.057
Nodal-Status (N)	1.12	0.89–1.40	0.343	1.16	0.94–1.43	0.176

ECOG, ECOG performance status; Charlson, Charlson Comorbidity Index; PCI, prophylactical cranial irradiation; EP vs. CP, cisplatin/etoposide vs. carboplatin/etoposide; CTV, clinical target volume (≥median CTV); RCT, radio-chemotherapy. Bold font indicates statistical significance (*p* < 0.05)

**Table 4 cancers-16-01953-t004:** Multivariable Cox proportional hazard-censored regression analysis of factors associated with overall survival and progression-free survival in patients with limited disease small cell lung cancer.

Overall Survival			
Variable	HR	95%-CI for HR	*p* Value
Sex	0.49	0.29–0.80	**0.004**
PCI	0.27	0.132–0.54	**<0.001**
Simultaneous RCT	0.43	0.244–0.77	**0.004**
EP vs. CP	0.57	0.319–0.99	**0.049**
**Progression-Free Survival**			
**Variable**	**HR**	**95%-CI for HR**	***p* Value**
Simultaneous RCT	0.66	0.38–1.12	0.125
EP vs. CP	0.50	0.29–0.86	**0.013**

PCI, prophylactical cranial irradiation; RCT, radio- and chemotherapy; EP vs. CP, Cisplatin/Etoposide vs. Carboplatin/Etoposide. Bold font indicates statistical significance (*p* < 0.05).

## Data Availability

The data presented in this study are available upon reasonable request from the corresponding author.

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
