# Peer review of "Therapeutic Patterns and Clinical Outcomes in Limited Disease Small Cell Lung Cancer: A Decade of Analysis at a Tertiary Cancer Center"

_cancers, 2024, doi:10.3390/cancers16111953_

Round 1

Reviewer 1 Report (Previous Reviewer 1)

Comments and Suggestions for Authors

Revised version is acceptable. No additional comments.

Author Response

Reviewer 2 Report (New Reviewer)

Comments and Suggestions for Authors

The paper provides a thorough examination of treatment patterns, outcomes, and survival predictors in LD-SCLC patients, offering significant insights and implications for future research and clinical practice. Structurally, it is well-organized, progressing logically from treatment patterns to survival predictors, limitations, strengths, and conclusions. The interpretation of results is commendable, particularly concerning the survival benefits associated with specific chemotherapy regimens and the role of PCI, with due acknowledgment of potential confounders and study limitations. However, to enhance its value further, it could delve deeper into the clinical implications for oncologists and multidisciplinary teams managing LD-SCLC patients. Exploring the molecular mechanisms underlying the observed differences in PFS and OS between CP and EP treatments could also be beneficial. Additionally, discussing potential alternative strategies to PCI and exploring novel treatment modalities in more detail would enrich future research avenues. Overall, refining the clarity and impact of the study's conclusions can make this discussion a valuable resource for both researchers and clinicians involved in LD-SCLC management.

Author Response

Reviewer 3 Report (New Reviewer)

Comments and Suggestions for Authors

The article tries to highlight some possible prognostic characteristics associated with a better clinical outcome in patients affected by LD-SCLC treated with the standard of care (chemotherapy and radiotherapy).

Authors do conclude that the use of cisplatin rather than carboplatin, the concomitant treatment rather than sequential, the use of PCI rather than the simple follow up, and the gender are prognostic factors, but these clinical characteristics have already been explored and are known validated prognostic factors.

All these things considered, I do not find any novelties in the article. 

Comments on the Quality of English Language

Several fluency issues should be addressed. Some sentences are not completed. A major revision is needed.

Author Response

This manuscript is a resubmission of an earlier submission. The following is a list of the peer review reports and author responses from that submission.

Round 1

Reviewer 1 Report

Comments and Suggestions for Authors

This study may potentially generate a hypothesis to evaluate the outcome of two chemo regimens, provided the two groups are at least somewhat balanced. In this retrospective study, groups are imbalanced in several clinical and therapeutic aspects. Specifically, 62 and 31 patients are in the cisplatinum and caroplatinum groups, respectively. Further, 73% vs. 25% received concurrent chemotherapy in the two groups, respectively. It is unknown at what cycle of chemotherapy the radiation was added. The dose fraction employed (table 1) is significantly imbalanced between the two groups and missing details. There are no details of PCI. There is no definition of very limited stage. 

The results section text could be clearer and more precise, and the discussion needs to be more adequate relating to the specific outcomes of the study. I am not sure how the study outcomes relate to personalized medicine as outlined by the authors.?

The better outcome of cisplatinum/etoposide is likely due to concurrent chemo rather than the type of chemotherapy. The current study is underpowered for any conclusion or to generate a hypothesis for a randomized controlled trial and does not add anything new to the existing literature on the management of limited-stage small-cell lung cancer. 

Comments on the Quality of English Language

As above.

Reviewer 2 Report

Comments and Suggestions for Authors

Dear Authors,

Thanks for your efforts in SCLC patients. You reported that female patients had better OS, and the type of chemotherapy and PCI remained independent predictors of survival. I have some questions.

1. The information on surgery was not clear.

2. The multivariable analysis did not include the TNM staging system.

Reviewer 3 Report

Comments and Suggestions for Authors

The authors aim to look at their institutional experience tackling the 'gray' areas of managing Limited Stage Small Cell Lung Cancer, where modern data is conflicting and there is no "right" answer per guidelines, societies, and sometimes within an individual's practice.  The two main questions they focus on are the use of Prophylactic Cranial Irradiation (PCI) and type of systemic therapy (EP vs CP).  In 93 patients (out of 210 institutional LS-SCLC) they found worse OS with carboplatin regimens as well as better OS with PCI. As they acknowledge this is a retrospective study and so hard to extrapolate to treatment recommendations.  This is especially true due to discrepancies in performance status.  Critically, the authors exclude patients who switched chemotherapy.  Most interesting in real-world data is the ability to see some of these issues.  For example, would a patient who was initially supposed to get Cis Etop--but failed to complete it due to toxicity--do better than someone who got Carbo Etop? The exclusion criteria here limit some of the findings.  Furthermore, we don't know at all how the PCI groups are balanced (or not).  Finally, in a study of treatment patterns, they don't look at timing of RT with cycle of chemo (e.g. 1 vs 2 vs completely sequential Tx) or at dosing of thoracic radiation (twice daily, 60 Gy, 66 Gy, 70 Gy) which continues to be a subject of debate. Overall this is an important and ongoing question with trial data limited by use of patients that are more young and better performance status than typical patients.  However, the selection criteria used to create this cohort also appears to be non-representative of "real world" experience. Notably they exclude >50% of their own LS-SCLC with criteria that likely exclude the patients we need to know about and for whom these questions are most salient. This as well as other remarks below suggest need for major revision. 

 - In abstract please include details about pts (e.g. how many included in analysis, median f/u, and perhaps some demographics like median age and % with ECOG 0)

 - I don't understand the definition of Very limited Dz vs Limited Dz.  This isn't a typical definition I have seen before.  Some people use Non-Small Cell Lung Cancer Staging (e.g. Stage I-II vs Stage III) which could be helpful

 - Additionally a stratification by disease volume (e.g. CTV, GTV, and/or PTV size) could be helpful at mitigating bias from a known confounder

 - As above, excluding change of systemic therapy regimen prevents us from seeing patients who were thought to be fit enough for Cis Etop but may in fact have a detriment to survival if treated with more aggressive chemo.  This confounds the ultimate finding that Cis Etop is better by effectively excluding all the patients who had the worst reaction to Cis Etop.  I could be wrong, but without the data we won't know

 - Make clear that the <=50 and >50 Gy in Table 1 are referring to once daily radiation regimens

 - It would be nice to have consistency with performance status / co-morbidities. I think either analyze ECOG and Charlson Co-morbidity with a binary (e.g. ECOG 0 or ECOG 0-1 vs other and CCI 0-1 or 0-2 vs other) or into more complete groups.  Many trials and other studies use ECOG 0-1 as "good" performance status.  That said there's already a significant difference in ECOG 0 so could break down ECOG 0, 1, 2+ and CCI 0, 1, 2, 3+

 - Even though not significant, there does seem to be clear trend to worse CCI in Carbo Etop Arm.  I think if you bundled them as discussed above (e.g. CCI 0-1 or 0-2 vs other) you might have statistics validate the 'eye test'.

 - Usually I'm used to defining timing of radiotherapy relative to chemo (e.g. RT started Cycle 1 vs 2 vs later) not opposite as described here --> recommend reformat as # (%) pts treated with RT at Cycle 1 vs 2 vs 3 vs 4 vs later

 - I don't see follow-up time (median, range, IQR, anything) in the publication. Hard to interpret median OS of 48 months for concurrent chemo when only 17 of initial 93 patients evaluable at 48 months.